# Niosomal Formulation of a Lipoyl-Carnosine Derivative Targeting TRPA1 Channels in Brain

**DOI:** 10.3390/pharmaceutics11120669

**Published:** 2019-12-10

**Authors:** Francesca Maestrelli, Elisa Landucci, Enrico De Luca, Giulia Nerli, Maria Camilla Bergonzi, Vieri Piazzini, Domenico E. Pellegrini-Giampietro, Francesca Gullo, Andrea Becchetti, Francesco Tadini-Buoninsegni, Oscar Francesconi, Cristina Nativi

**Affiliations:** 1Department of Chemistry “Ugo Schiff”, University of Florence, 50019 Florence, Italy; enrico.deluca1187@gmail.com (E.D.L.); giulia.nerli@unifi.it (G.N.); mc.bergonzi@unifi.it (M.C.B.); vieri.piazzini@unifi.it (V.P.); francesco.tadini@unifi.it (F.T.-B.); cristina.nativi@unifi.it (C.N.); 2Department of Health Sciences, Clinical Pharmacology and Oncology Unit, University of Florence, 50139 Florence, Italy; elisa.landucci@unifi.it (E.L.); domenico.pellegrini@unifi.it (D.E.P.-G.); 3Department of Biotechnologies and Biosciences, University of Milano-Bicocca, 20126 Milan, Italy; francesca.gullo@unimib.it (F.G.); andrea.becchetti@unimib.it (A.B.)

**Keywords:** TRPA1 antagonist, cortical spreading depression, blood brain barrier, niosomes

## Abstract

The transient receptor potential akyrin type-1 (TRPA1) is a non-selective cation channel playing a pivotal role in pain sensation and neurogenic inflammation. TRPA1 channels expressed in the central nervous system (CNS) have a critical role in the modulation of cortical spreading depression (CSD), which is a key pathophysiological basis of migraine pain. ADM_09 is a recently developed lipoic acid-based TRPA1 antagonist that is able to revert oxaliplatin-induced neuropathic pain and inflammatory trigeminal allodynia. In this context, aiming at developing drugs that are able to target TRPA1 channels in the CNS and promote an antioxidant effect, permeability across the blood–brain barrier (BBB) represents a central issue. Niosomes are nanovesicles that can be functionalized with specific ligands selectively recognized by transporters expressed on the BBB. In this work, the activity of ADM_09 on neocortex cultures was studied, and an efficient formulation to cross the BBB was developed with the aim of increasing the concentration of ADM_09 into the brain and selectively delivering it to the CNS rapidly after parenteral administration.

## 1. Introduction

The transient receptor potential akyrin type-1 (TRPA1) is a non-selective cation channel playing a pivotal role in pain sensation and neurogenic inflammation [1]. TRPA1 is prominently present in somatosensory neurons where it is reported to detect a great deal of external stimuli, including mechanical stress, temperature, and environmental irritants [2]. Since it plays a crucial role in the generation and maintenance of inflammatory pain, TRPA1 has become an attractive target for anti-inflammatory and analgesic therapies [3]. Although studies have been mainly focused on peripheral TRPA1 channels, they are also expressed in the central nervous system (CNS) [4] and in particular in cortical neurons, hippocampal pyramidal neurons, and astrocytes [5]. Recent studies have highlighted the critical role of central TRPA1 channels in the modulation of cortical spreading depression (CSD) [6], which is a transient propagating excitation of synaptic activity followed by depression. CSD is a key pathophysiological basis of migraine pain [7] and triggers migraine-like behavior through the generation of neuroinflammatory responses in rats [8]. The modulation of CSD by TRPA1 activation takes place by stimulating the release of calcitonin gene-related peptide (CGRP), which increases cortical susceptibility to CSD [9]. Considering that TRPA1 channels are sensitive to oxidative stress, reactive oxygen species (ROS) have been recently proposed as an activator of central TRPA1 channels, triggering the release of CGRP and consequently the development of migraines [5]. Therefore, the inhibition of ROS and deactivation of TRPA1 channels may have a synergistic therapeutic benefit in preventing stress-triggered migraine via CGRP by a central mechanism. In this context, when developing drugs that are able to target TRPA1 channels in the CNS and promote an antioxidant effect, permeability across the blood–brain barrier (BBB) represents a central issue.

The BBB represents a diffusion barrier that is essential for protecting the CNS from toxic substances. Most compounds are not allowed to cross this barrier, and most drugs cannot reach the brain. Only small, lipophilic molecules and compounds recognized by specific transporters on the barrier can passively cross the BBB. An innovative strategy to selectively deliver drugs to the brain is the use of nanocarriers such as liposomes, nanoparticles, and niosomes [10]. Niosomes are nanovesicles made of synthetic surfactants; these are more stable and less expensive than liposomes and devoid of toxicity, with an inner aqueous core where hydrophilic drugs can be loaded [11,12]. For selective brain targeting, these nanocarriers can be functionalized: on the outer surface, specific ligands can be inserted to be recognized by transporters expressed on the BBB [13,14]. *N*-palmitoyl-glucosamine (NPG) already proved to enhance the concentration in the brain of a drug delivered by means of colloidal systems probably by interaction with the glucose transporter GLUT-1 [15,16].

In recent years, our group has developed a family of lipoic acid-based TRPA1 antagonists named ADM, which are able to revert not only oxaliplatin-induced neuropathic pain [17,18,19] but also inflammatory trigeminal allodynia [20], which is particularly relevant in the treatment of orofacial pain [21] and migraine pain [22]. In detail, the progenitor ADM_09 (Figure 1), obtained condensing commercially available (±) α-lipoic acid with l-carnosine, is a hydrophilic ionic small molecule exerting a Ca^2+^-mediated TRPA1 antagonism. Interestingly, ADM_09 did not modify normal behavior in rats and did not show any toxicity toward astrocyte cell viability, nor any relevant cardiotoxicity. Moreover, the antioxidant free radical scavenger properties of ADM_09 were highlighted *ex-vivo* (superoxide anion levels in rat cortical astrocytes) as well as *in-vivo* (reversion of oxaliplatin-induced, ROS-mediated, neuropathic pain) [17]. The dual activity of ADM_09 (TRPA1 antagonist and antioxidant) is potentially paradigmatic for reducing the cortical susceptibility to CSD. However, despite the good results obtained in the modulation of peripheral TRPA1 channels, the activity of ADM_09 on CNS-expressed TRPA1 remains still unexplored.

In this work, the activity of ADM_09 on neocortex cultures was studied, and an efficient formulation to cross the BBB was developed with the aim of increasing the concentration of ADM_09 into the brain and selectively delivering it to the CNS rapidly after parenteral administration.

## 2. Materials

Solulan™ C24 (SOL) (Poly-24-oxyethylenecholesteryl ether) was a gift by Lubrizol (Cleveland, OH, USA). Cholesterol (CH), cholesterol hemisuccinate (CHE), sorbitan monopalmitate (Span 40, HLB 9.8), sorbitan stearate (Span 60, HLB 4.7), palmitic acid *N*-hydroxysuccinimide, and glucosamine, triton X-100 (TX) were purchased from Sigma-Aldrich (Milan, Italy), while hydroxypropyl-beta-cyclodextrin (HPβCD) was kindly donated by Roquette (Alessandria, Italy). All the other reagents were of analytical grade. 

*N*-palmitoyl glucosamine (NPG) was synthesized according to Bragagni et al. 2012 [15], and carefully characterized as previously described [16]. ADM_09 was synthesized according to a previous method [17]. All other reagents were of analytical grade.

## 3. Methods

### 3.1. Neuronal Activity Tests

Primary neocortex cultures (without the hippocampus) were prepared from brains extracted from neonatal (P1–P3) mice (Envigo, Italy), as previously described [23]. Mice were handled and treated according to the Principles of Laboratory Animal Care (directive 86/609/EEC), as endorsed by the Italian Ministry of Health. Cells were plated at densities of 600/900 × 10^3^ cells/mL on multi-electrode array (MEA) dishes precoated with polyethyleneimine 0.1%. After 3 h of incubation, the plating medium was replaced by neurobasal medium supplemented with B27 (Thermo Fisher Scientific, Monza, Italy, glutamine 1 mM, and basic fibroblast growth factor 10 ng/mL. Subsequently, cultures were kept at 37 °C, in 5% CO_2_, and covered with gas-permeable covers (MEA-MEM; ALA Scientific Instruments, Farmingdale, NY, USA) throughout the culture period (15 days *in-vitro*, DIV). Half of the medium was substituted every 3 days, and on the day preceding the drug application. For MEA recording, we used arrays formed by 60 electrodes (indium tin oxide electrodes with 30 μm diameter, spaced 200 μm apart; MultiChannel Systems, Reutlingen, Germany). Each dish had a recording area of ~2.7 mm^2^, containing on average of 8–9000 cells.

Data were analyzed as previously reported [24,25]. For the sake of brevity, we only report here the spike rate (SR; in Hz), the normalized excitability (spikes per neuron), and the intervals between network bursts (network burst inter-burst intervals, NB-IBI). These parameters provide an overall view of the network excitability and were calculated for consecutive intervals in the different experimental conditions. The above parameters were calculated for clusters of putative excitatory and inhibitory neurons and distinguished on the basis of firing parameters, as previously reported [25]. ADM_09 was dissolved in distilled water, and the solution was applied in the bath. The injected volume was always lower than 1% of the culture solution volume. The network activity was measured for 15 min at each concentration.

### 3.2. Preparation of Niosomes

Niosomes were prepared by three different methods: thin layer evaporation–paddle, thin layer evaporation–vortex, and thin layer evaporation–frozen and thawed.

The thin layer evaporation–paddle (TLE-P) stirring technique was a partial modification of a previous method [15]. Briefly, the lipidic phase was dissolved in CHCl_3_ and completely removed by rotary evaporation under vacuum to form a thin layer on the flask wall. Then, the layer was hydrated with 20 mL of hydrophilic phase under stirring by a paddle at 2000 rpm for 30 min, heating in a water bath at 65 °C.

The thin layer evaporation–vortex (TLE-V) stirring technique was also used; after the thin layer hydration, the suspension was submitted to 4 cycles of heating in a water bath (3’, 65 °C) and vortex agitation (3’, 20 Hz).

The thin layer evaporation–frozen and thawed (TLE-F) technique was used next; after the thin layer hydration, the suspension was submitted to 4 cycles of heating in a water bath (3’, 65 °C) and vortex agitation (3’, 20 Hz) and 4 cycles of freezing (3’ in liquid N_2_) and heating in a water bath (3’, 65 °C).

Niosomes were prepared using different concentrations of the lipid components: Span 40 or 60 as the main surfactant, CH or CHE as a stabilizer, and SOL. NPG concentration was maintained constant since, as observed in a previous study, an increase produces a drug crystallization that reduces the formulation stability [15]. The total amount of lipid components was 19.06 mg/mL, ADM_09 was added in the hydrophilic phase (PBS pH 7.4) at the final concentration of 1.4 mg/mL, and when added to the formulation, HPβCD was added with ADM_09 at a 1:1 drug to cyclodextrins molar ratio.

All the suspensions were centrifuged at 4000 rpm for 15 min (HERMLE Z 200 A centrifuge) according to a previous method [12]. The supernatant was collected, and 10 mL were subjected to sonication in an ice bath with a Sonoplus HD2200 ultrasound homogenizer (Bandelin Electronic GmbH, Berlin, Germany) equipped with a KE76 probe at 50% power for 5 min for 2 cycles.

### 3.3. Analytical Method for ADM_09 Determination

The HPLC method for ADM_09 quantification was developed according to two different methods reported in the literature for lipoic acid and carnosine [26,27] opportunely modified. The quantitative assay was carried out with a Merck Hitachi Elite La Chrom equipped UV-VIS detector settled at 260 nm. The column was an Agilent Zorbax CN (4.6 × 150 mm, 5 µm). The mobile phase was acetonitrile to ammonium acetate pH = 4.6 15:85 v/v, the volume injected was 20 µL, the flux was 0.6 mL/min, and the oven temperature was 40 °C. The standard curve was prepared by dissolving ADM_09 in PBS pH 7.4 The linearity was determined on five concentration levels (from 0.1 to 0.5 mg/mL) with three injections for each level. The coefficient of linear correlation was above 0.999. LOQ = 0.0192 mg/mL, LOD = 0.0064 mg/mL.

### 3.4. Niosomes Characterization

Particle size, polydispersion index (PDI), and the ζ-potential of niosomes were measured by dynamic light scattering using a Zetasizer Nanoseries ZS90 (Malvern Instrument, Worcestershire, UK) at an angle of 90 in 0.01 m width cells at 25 ± 1 °C. Colloidal suspensions were properly diluted with distilled water in order to avoid scattering phenomena. Each sample was analyzed in triplicate. The morphology was evaluated using a transmission electron microscope (CM12 TEM, PHILIPS, Amsterdam, The Netherlands), which was equipped with an OLYMPUS Megaview G2 camera (Segrate, Italy). The samples were analyzed at 80 kV. For TEM analysis, a drop of the vesicle dispersion was placed onto a Cu/C grid and stained by adding a drop of 2% (w/w) uranyl acetate solution; the excess solution was removed using filter paper, followed by a thorough air-drying.

### 3.5. Determination of Drug Entrapment

Drug entrapment was determined by the dialysis bag method. The membrane (Spectra/Por^®^, MW cut-off 12000, Spectrum (New Brunswick, NJ, USA) was filled with 1 mL of the samples and kept for 24 h in 5 mL of PBS pH 7.4 to remove the free ADM_09. A 0.5 mL aliquot of niosomes suspension was withdrawn, diluted with 0.5 mL of a 10% *w*/*v* solution of Triton X-100 to destroy the niosomes, and analyzed by HPLC UV/VIS as previously described. Drug entrapment has been calculated as drug loading (DL%) and entrapment efficiency (EE%) according to Equations (1) and (2):
DL% = mg entrapped drug/mg niosomes × 100(1)

EE% = concentration of entrapped drug/total drug concentration × 100(2)

### 3.6. StabilityStudies

Stability studies were conducted on ADM_09 solutions. Solution of ADM_09 in PBS pH 7.4 at the concentration of 0.6 mg/mL were placed in glass vials and amber-glass vials and stocked at 5 ± 3 °C, 25 °C at 60% of relative humidity (RH), and 37 °C at 75% of RH using a constant climatic chamber (HPP110, Memmert Gmb, Schwabach, Germany) according to the International Council for Harmonisation of Technical Requirements for Pharmaceuticals for Human Use (ICH) guidelines for pharmaceutical products stability [28]. The residual ADM_09 concentration was evaluated during 4 weeks using HPLC-UV/VIS. The physical stability of niosomal suspensions was evaluated during 8 weeks in the same conditions. The colloidal suspensions were analyzed in terms of mean particle size, PDI, and ζ-potential. Visual inspection was also employed to detect any crystallization process or mold formation. Stability studies in fetal bovine serum (FBS) were conducted over 24 h at 37 °C in order to evidence the colloidal system stability in such medium.

### 3.7. In-Vitro Release Studies

Drug release *in-vitro* was assessed using the dialysis method. The niosomial suspension or a solution containing 1.4 mg/mL of drug was placed in a cellulose acetate dialysis bag (Spectra/Por®, MW cut-off 12000, Spectrum, New Brunswick, NJ, USA), immersed in PBS pH = 7.4, and incubated at 37 °C under electromagnetical stirring. At predetermined time points (30, 60, 90, 120, 150, and 180 min), a volume of receptor solution was withdrawn and refilled with an equivalent volume of fresh PBS, and the concentration of ADM_09 was determined using HPLC-UV/VIS.

### 3.8. hCMEC/D3 Cell Culture

The immortalized human cerebral microvascular endothelial cell line, hCMEC/D3 (Millipore Cat. # SCC066), belongs to human temporal lobe micro vessels isolated from tissue excised during surgery for epilepsy control. Cells were seeded in a concentration of 2.5 × 10^4^ cells/cm^2^ and grown at 37 °C in an atmosphere of 5% CO_2_ in 25 cm^2^ rat tail collagen type I coated culture flasks [29]. The media (EndoGRO^TM^- MV Complete Media Kit supplemented with 1 ng/mL FGF-2) was changed every three days, and cells were grown until they were 90% confluent. Cells were passaged at least twice before use.

### 3.9. MTT Assay

To assess cell viability after ADM_09, niosomes (NIO), and loaded niosomes (NIO ADM_09) exposure, an 3-(4,5-dimethylthiazol-2-yl)-2,5-diphenyltetrazolium bromide (MTT) assay was performed [30]. Cells were seeded in a 24-well plate (6 × 104 cells/cm2) precoated with Collagen Type I, Rat Tail and grown at 37 °C in an atmosphere of 5% CO2 in EndoGROTM Basal Medium (EBM-2). Then, the cells were incubated, approximately at 70–80% confluency, with different concentrations of ADM_09 (0.14 and 0.07 mg/mL), NIO, and NIO + ADM_09 (0.14 and 0.07 mg/mL) obtained by dilution (1:10 and 1:20) of the formulation in EBM-2 for 2 and 24 h. For lactate dehydrogenase (LDH) assay, the medium of each well was separated from the cells and stored. The cells were treated with 1 mg/mL of MTT for 1 h at 37 °C; 5% CO2 and DMSO was added to dissolve MTT formation, and absorbance was measured at 550 and 690 nm. Cell viability was expressed as a percentage compared to the cells incubated only with EBM-2 (positive control). TX was used in the MTT assay as the negative control, since its detergent action disrupts the cells.

### 3.10. LDH Assay

LDH assay was performed in order to test the cytotoxicity after ADM_09 and niosomes exposure [31]. The medium resulting from the incubation of ADM_09 and niosomes with cells was centrifuged at 250 g for 10 min at room temperature, and the supernatant was separated from the deposited cells in each well. The release of LDH into culture supernatants was detected by adding catalyst and dye solutions of a Cytotoxicity Detection Kit (LDH) (Roche Diagnostics, Indianapolis, USA). The absorbance values were recorded at 490 nm and 690 nm. Cytotoxicity was expressed as a percentage compared to the maximum LDH release in the presence of TX (positive control). EBM-2 was used as negative control, since no cytotoxicity was detected in such conditions [30].

### 3.11. hCMEC/D3 Cell Culture for Transwell Permeability Studies

Permeability studies were conducted in order to evidence the drug permeation of ADM_09 alone or formulated as niosomes, across the hCMEC/D3 monolayer. With this aim, high density pore (2 × 106 pores/cm^2^) transparent polyethylene terephthalate (PET) membrane filter inserts (0.4 μm, 23.1 mm diameter, Falcon, Corning BV, Amsterdam, Netherlands) were coated as previous described in [29] and used in 6-well cell culture plates for all transcytosis assays. Apical and basolateral chambers contain, as optimal media volume, respectively, 1 and 1.2 mL. hCMEC/D3 cells were seeded onto the apical side of the Transwell inserts at a density of 6 × 104 cells/cm^2^ in 1 mL assay media, after a calibration with assay medium for 1 h. Then, 1.2 mL of fresh medium was added to the basolateral chamber. The assay medium was changed every 3 days, and for 7 days, cells were grown to confluence. The integrity of monolayer cells was confirmed also by the observation of cultures under phase-contrast microscopy or under bright-field optics using transparent membranes. Moreover, as an integrity control marker, the apparent permeability of fluorescein sodium salt (NaF) at a concentration of 10 µg/mL was evaluated [29]. For permeability studies, ADM_09 (0.14 and 0.07 mg/mL), NIO, and NIO + ADM_09 (0.14 and 0.07 mg/mL) obtained by 1:10 and 1:20 dilution of the formulation in EBM-2 were tested and incubated for 0.5, 1, and 2h in the apical donor compartment. At the end of the incubation, the amount of NaF and ADM_09 were quantified both in apical and basolateral compartments by HPLC-FLD [30] or HPLC UV/VIS, respectively. In the case of the formulation, EBM-2 was diluted with methanol, placed in the ultrasonic bath for 30 min, and then ultra centrifuged for 1 h at 11330 × g (4 °C). The apparent permeability coefficients (Papp) of free ADM_09 and ADM_09 encapsulated in NIO were calculated according to Equation (3) [29]:
P_app_ (cm/s) = V_D_/(A M_D_) × (ΔM_R_/Δt)(3)
where V_D_ = apical (donor) volume (cm^3^), M_D_ = apical (donor) amount (mol), and ΔM_R_/Δt = change in amount (mol) of compound in receiver compartment over time.

The recovery for ADM_09 and NaF was calculated according to Equation (4):
Recovery (%) = C_Df_V_D_ + C_Rf_V_R_/(C_D0_V_D_) × 100(4)
where C_Df_ and C_Rf_ are the final compound concentrations in the donor and receiver compartments, C_D0_ is the initial concentration in the donor compartment, and V_D_ and V_R_ are the volumes in the donor and receiver compartments, respectively. All the experiments were performed at least in triplicate.

### 3.12. Statistical Analysis

One-way analysis of variance (ANOVA) followed by the Student–Newman–Keuls multiple comparison post-test was performed on the data of niosomes characterization studies. All experiments were repeated at least three times, and results are expressed as a mean ± standard deviation.

Statistical significance of hCMEC/D cell viability and cellular uptake was analyzed using one-way ANOVA followed by the post hoc Tukey’s w-test for multiple comparisons. All statistical calculations were performed using GRAPH-PAD PRISM v. 5 for Windows (GraphPad Software, San Diego, CA, USA). A probability value (*p*) < 0.05 was considered significant.

## 4. Results and Discussion

### 4.1. Neuronal Tests

TRP channels are increasingly recognized as important regulators of brain function [32,33]. TRPA1 channels have been recently found to be expressed in the neocortex and to regulate network activity [4,5]. In particular, TRPA1 activation produces depolarizing effects in pyramidal neurons [4], and it is thus expected to increase excitability, at least during transient activation. As previously mentioned, ADM_09, along with its anti-ROS properties, presents TRPA1 inhibition properties. Hence, we treated our neocortex cultures (15 DIV) with this compound, at concentrations between 1 µM (0.41 µg/mL) and 100 µM (41.4 µg/mL). The results of a representative experiment are illustrated in Figure 2, in which increasing concentrations were applied at 1 h intervals, while the network activity was continuously registered.

As expected, ADM_09 decreased neuronal activity, which was reflected in a lower excitability (spikes per neuron; Figure 2A). The maximal effect was observed at 30 µM (12.4 µg/mL) ADM_09, with an IC_50_ of approximately 3 µM (1.24 µg/mL), and was more pronounced on the neuron cluster containing inhibitory neurons (black circles). This was presumably the cause of the partial network disinhibition produced by ADM_09, which decreased the intervals between network bursts (NB-IBI; Figure 2B), suggesting a higher network propensity to undergo a global firing burst. Such interpretation is consistent with the observation that the average decrease of the overall spike rate (SR; Figure 2C) was statistically significant in the inhibitory neurons’ cluster. The easiest interpretation of these effects is that partially inhibiting pyramidal cells with ADM_09 have comparatively higher effects on the strong excitatory connections between pyramidal neurons and fast-spiking GABAergic cells [34], thus resulting in network disinhibition. Regardless of the specific mechanism, our results showed that ADM_09 produces a rapid direct effect on neuronal network excitability, and that the effect is promptly reversible on washout.

### 4.2. Formulation of ADM_09

ADM_09 was formulated into NPG-functionalized niosomes with the aim of driving this amphiphilic molecule into the brain and rapidly targeting the CNS after parenteral administration. Three different preparation methods were tested, in order to evidence the most suitable method to obtain niosomes for parenteral administration. The requested characteristics were a particle size lower than 200 nm, a good stability in term of ζ-potential, and a PDI ≤ 0.2. In this first phase, the experiments were conducted on empty niosomes with a composition of Span 60/CH/SOL/NPG 5.2:3.6:3.6:1 w/w ratio. The results obtained with the three methods are reported in Table 1.

Among the different methods, TLE-V can be discarded, since the stirring by the vortex is not able to produce niosomes with a size lower than 200 nm and a PDI ≤ 0.2. The other two methods TLE-P and TLE-F did not show differences in terms of particle size, PDI, and ζ-potential. Then, TLE-P was selected for its greater simplicity. In fact, while in the TLE-F method, after rehydration and vortex agitation, the sample should be submitted to cycles of freezing and heating; the TLE-P the paddle agitation is able to obtain the same results in a single step.

Once the preparation method is selected, niosomes were prepared at different components ratios and compositions, as reported in Table 2, in order to evidence the effect of colloidal system composition. The NPG amount was maintained constant due to its role of brain-targeting agent and since, as observed in a previous study, an increase produces a drug crystallization that reduces the formulation stability [15]. Span 40 and CHE were selected because of their higher hydrophilicity with respect to Span 60 and CH, with the aim to improve the entrapment of a hydrophilic component such as ADM_09. HPβCD was selected with the same purpose, since recent studies reported a possible interaction between cyclodextrins and R-α-lipoic acid [35,36] and a positive effect of the cyclodextrins on hydrophilic drug entrapment in niosomes [37]. As shown in Table 2, modifying the lipidic phase ratio (batches 1, 2, and 3), the particle size and PDI worsen, showing that the components ratio in the formulation cannot be varied without changing the niosomes’ characteristics.

The substitution of Span 60 with Span 40 (batch 4) and of CH with CHE (batch 5) produced particles with a dimension slightly higher than 200 nm, while the simultaneous substitution of Span 60 with Span 40 and of CH with CHE (batch 6) and the addition of HPβCD (batch 7) produced vesicles with the suitable characteristics. Then, these batches were selected for ADM_09 entrapment, and the results are reported in Table 3.

ADM_09 entrapment produced an increase in particle size due to the drug entrapment in the hydrophilic phase that did not affect the particle surface charge (ζ-potential remained unchanged). The hydrophilic components that were introduced in order to increase the drug entrapment, Span 40 and CHE, unexpectedly reduced the drug entrapment. In addition, the attempt to improve drug entrapment using a cyclodextrin (HPβCD) did not produce the desired effect, even if in the literature a positive effect of cyclodextrins on hydrophilic drug entrapment in niosomes was reported by Machado et al. 2018 [37].On the other hand, Machado et al. observed a better effect of βCD on the entrapment of a hydrophilic dye in niosomes with respect to its derivative modified amphiphilic HPβCD; however, for our application, it is not possible to change the HPβCD (FDA-approved for a parenteral use) with a more lipophilic compound, such as the natural βCD, because of its parenteral toxicity. The selected composition was then Span 60, 8 mg/mL; CH, 5.73 mg/mL; SOL, 5.33 mg/mL; NPG, 1.58 mg/mL; and ADM_09, 1.4 mg/mL.

Niosomes morphology was observed with TEM analysis, and the results are reported in Figure 3. In Figure 3A,C, the micrographs of NIO are reported, while in Figure 3B,D, the micrographs of NIO ADM_09 are shown at the same magnification. The vesicles present a spherical shape, with a particle size compatible with the DLS findings. A certain reduction in size is due to the nanoparticles drying process during the sample preparation.

### 4.3. Stability Studies

ADM_09 solutions in PBS pH 7.4 at the initial concentration of 0.6 mg/mL were stocked in glass and amber-glass vials at 5, 25 and 37 °C at 60% RH. The resulting drug solutions were stable at 5 °C after 4 weeks, while a reduction was observed at 25 °C (0.5 mg/mL), and a more evident degradation was revealed in the solution stocked at 37 °C after 1 week (0.4 mg/mL). No difference was observed between glass and amber-glass vials, indicating that the drug degradation was due to the temperature and not to a light effect.

Stability studies were conducted for 8 weeks on NIO and NIO ADM_09. The colloidal suspensions were analyzed in terms of mean particle size, PDI, and ζ-potential. The results in terms of particle size are reported in Figure 4.

After drug entrapment, an increase of particle size was observed, which was due to drug inclusion in the colloidal systems. In the colloidal suspensions maintained at 5 °C (Figure 4A), a slight variation in particle size was observed. This variation is not considered meaningful, since it is not associated with any PDI and ζ-potential variation (data not shown). On the other hand, in the samples maintained at 25 °C and 37 °C, a reduction of ζ-potential and an increase of particle size (Figure 4B,C) and PDI were revealed. These findings are more evident in case of loaded NIO, indicate a certain aggregation and instability of the colloidal suspensions maintained at higher temperature, and are probably compounded also to drug degradation. As expected, since the preparation was carried out under non-sterile conditions and no preservative was added to the formulation, mold formation was observed in the samples stocked at 25 °Cand 37 °C after 4 weeks.

Stability studies of niosomes were conducted on FBS at 37 °C in order to simulate the *in-vivo* behaviour, and the results are summarized in Table 4. We also observed a light increase of the PDI of niosomes with respect to the colloidal systems in PBS at 37 °C (*p* < 0.05), which was probably due to the protein presence, but the particle size was maintained during the first 3 h, thus confirming the niosomes’ stability in the medium.

### 4.4. In-Vitro Release Studies

*In-vitro* release studies of ADM_09 from the aqueous solution and the niosomal suspensions were conducted with the dialysis bag method at 37 °C, and the results obtained for the first 3 h are reported in Figure 5.

As shown, ADM_09 was completely released from the solution in the first 90 min; in the case of drug formulation, a rapid drug release of almost 50% of the drug was observed in the first 30 min, while a slower and controlled drug release was obtained during 3 h. This bimodal release is typical of hydrophilic drugs from vesicles systems: the burst effect is due to the fast release of the unentrapped drug and the drug released from the niosomes controlled by the diffusion according to Higuchi model (R^2^ 0.968). After 24 h, a complete drug release was evidenced.

### 4.5. MTT and LDH Assays

MTT and LDH assays were performed on the hCMEC/D3 cell line to evaluate the effect of ADM_09, NIO, and NIO + ADM_09 on cell viability and cytotoxicity, and permeability studies were also conducted in Transwell devices using the same cell line. The *in-vitro* cytotoxicity of the developed drugs was assessed by cell viability determination and membrane integrity evaluation using the hCMEC/D3 cell line in MTT and LDH assays, respectively (Figure 6).

When cells were exposed to different concentrations of ADM_09 (0.14 and 0.07 mg/mL), niosomes, and loaded niosomes (NIO ADM_09) for 2 h (Figure 6A) and 24 h (Figure 6B), no significant changes were observed in MTT metabolization or LDH release when compared to cells exposed to the EBM-2 medium alone. These results evidence that empty or ADM_09loaded niosomes did not affect the metabolic activity of the cells, and the drug did not alter the membrane integrity.

### 4.6. BBB Permeability Studies

The hCMEC/D3 brain microvascular endothelial cell line is a model of human BBB utilized to study the drug transport mechanisms [30,38]. The cells retain the expression of most transporters and receptors expressed *in-vivo* in the human BBB. The hCMEC/D3 apparent permeability coefficient (Papp) correlates well with *in-vivo* permeability data; therefore, permeability studies were performed to predict the permeability of free ADM_09 and NIO ADM_09 across the BBB. NaF was used as negative control, and its Papp was determined during all the transport experiments to monitor the integrity of the cell layer. The integrity of the cell layer was also assessed by phase-contrast microscopy [29]. The Papp of NaF was 4.13 ± 0.06 10^−6^ cm/s, and this value remained constant during the permeability assay. This result demonstrates the confluence of the monolayer and assesses the tight junction integrity [30]. Permeation studies on hCMEC/D3 cells showed an increased Papp of ADM_09 formulated in niosomial suspension compared to pure ADM_09 solution. The results, which are summarized in Figure 7, showed that the Papp of NIO ADM_09 after 30 and 60 min of experiments resulted in values higher than that of the free ADM_09, while after 60 min of incubation, the permeation was similar.

This finding revealed a rapid absorption of the drug from the cell layer, which was highly improved by niosomal formulation. Niosomal formulations composed by non-ionic tensides recently demonstrated their ability to improve brain uptake very quickly [39], and NPG-functionalized niosomal formulations were also reported to enhance the concentration in the brain of a delivered drug [15,16]. With this formulation, the drug that crosses the monolayer reaches a concentration significantly higher than the unformulated drug, and it is theoretically able to perform the pharmacological effect.

## 5. Conclusions

In this study, the activity of a TRPA1 antagonist, namely ADM_09, on neocortex cultures was studied, and an efficient formulation to cross the BBB was developed. The addition of ADM_09 to neocortex cultures showed a rapid decrease in neuronal activity that was restored only after washout, suggesting that an effective antagonism of TRPA1 channels partially inhibits pyramidal cells. ADM_09 formulation was carried out in NPG-containing niosomes. NPG was successfully employed for brain targeting of the vasoactive intestinal peptide, doxorubicin, and dynorphin-B [16,17,40]. Although the exact mechanism of action needs further investigation, the recognition of glucosamine exposed on NPG-bearing niosomes by the glucose transported GLUT-1, which was highly expressed on the BBB cell, has been hypothesized [40]. Among the different preparation methods, the TLE-P method proved to be simple, reproducible, and cost effective, resulting as a good and scale-up favorable process. The optimized formulation (Span 60 8 mg/mL CH 5.73 mg/mL SOL 5.33 mg/mL NPG 1.58 mg/mL) produced niosomes with suitable characteristics for parenteral administration with a good drug loading. NIO ADM_09 were found to be stable at 5 °C for 4 weeks, keeping dimensions below the intra-venous administration ideal diameter of 200 nm. Studies on hCMEC/D3 cell lines demonstrated the absence of toxicity and a good permeation enhancement of the nanosystems-formulated drug. Future experiments will be performed in order to obtain sterile formulations suitable to be tested in preclinical studies.

## Figures and Tables

**Figure 1 pharmaceutics-11-00669-f001:**
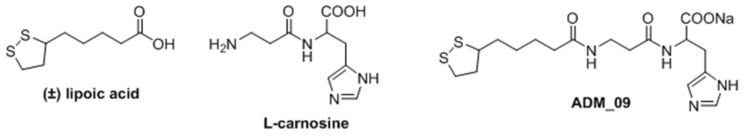
Structure of lipoic acid, carnosine, and ADM_09.

**Figure 2 pharmaceutics-11-00669-f002:**
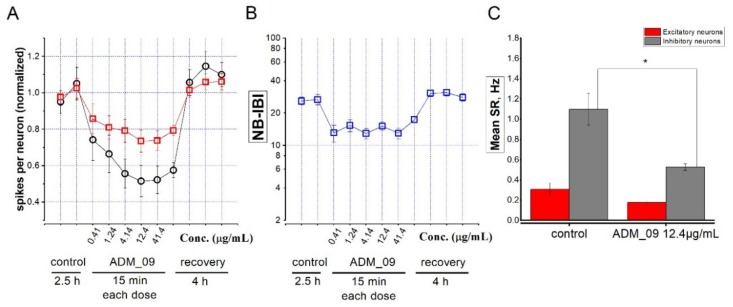
(**A**) Data points represent the average number of spikes per neuron, normalized to the average value measured in control extracellular solution (control), for 2.5 h. Next, ADM_09 was added at the indicated concentrations (µg/mL). Each concentration was left for 15 min. After the drug was removed (recovery), excitability returned to the control values within 1 h. Each data point for control and recovery represents 1 h recording, as indicated. Red squares: excitatory cluster; black circles: inhibitory cluster. (**B**) Same as A, but giving the average inter-burst interval between network bursts (NB-IBI). (**C**) Mean spike rate in control and ADM_09 (12.4 µg/mL) for the excitatory (red) and the inhibitory cluster (gray). * *p* < 0.05.

**Figure 3 pharmaceutics-11-00669-f003:**
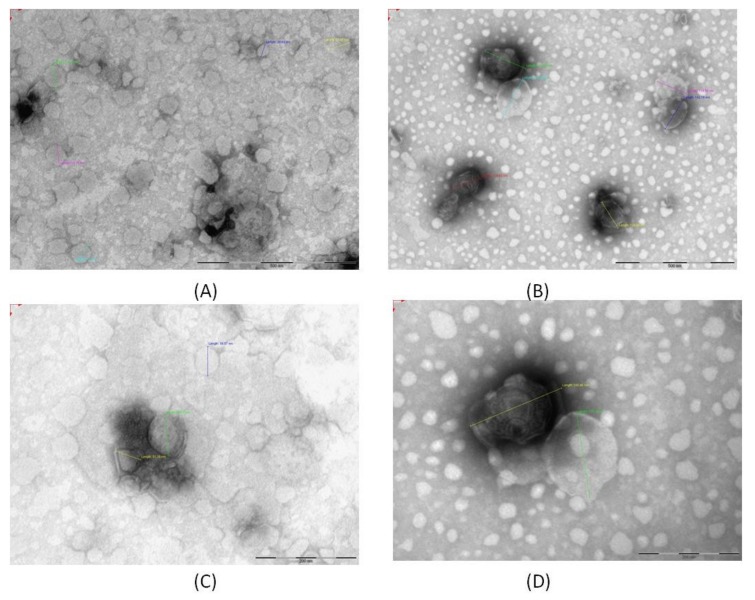
TEM micrographs of NIO (**A**) and NIO ADM_09 (**B**) and the corresponding magnification of NIO (**C**) and NIO ADM_09 (**D**).

**Figure 4 pharmaceutics-11-00669-f004:**
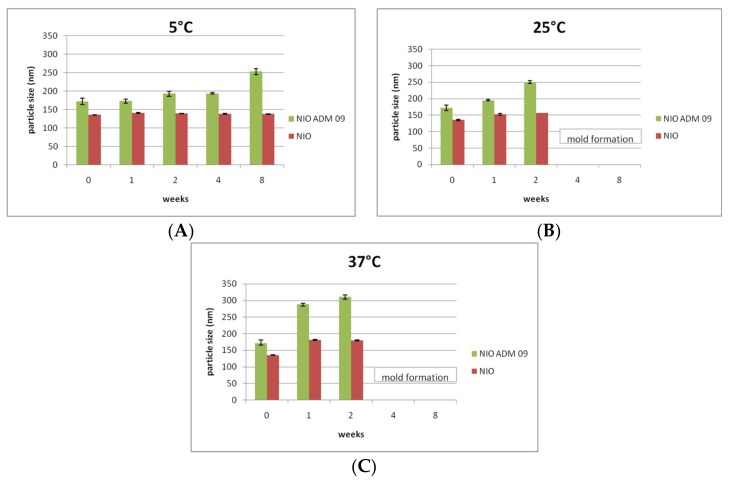
Particle size (nm) of NIO (green) and NIO ADM_09 (red) during the stability test conducted for 8 weeks at 5 °C (**A**), 25 °C (**B**), and 37 °C (**C**) (75% RH). RH: relative humidity.

**Figure 5 pharmaceutics-11-00669-f005:**
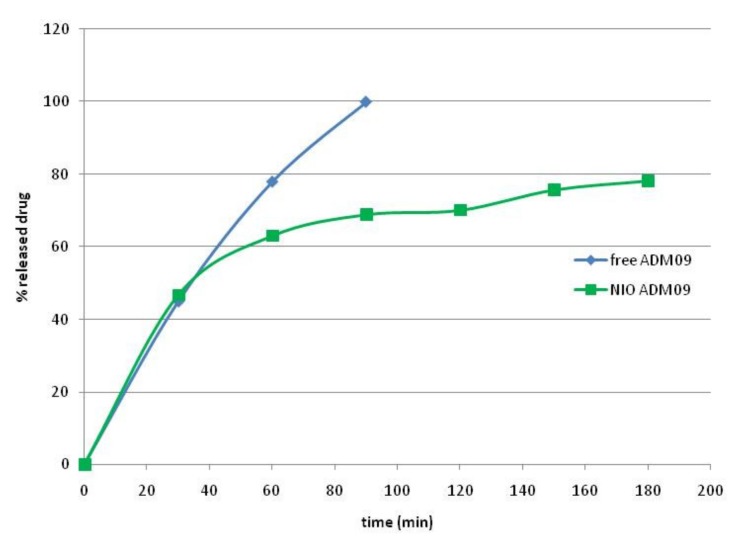
Drug release of ADM_09 from solution and loaded niosomes (NIO ADM_09).

**Figure 6 pharmaceutics-11-00669-f006:**
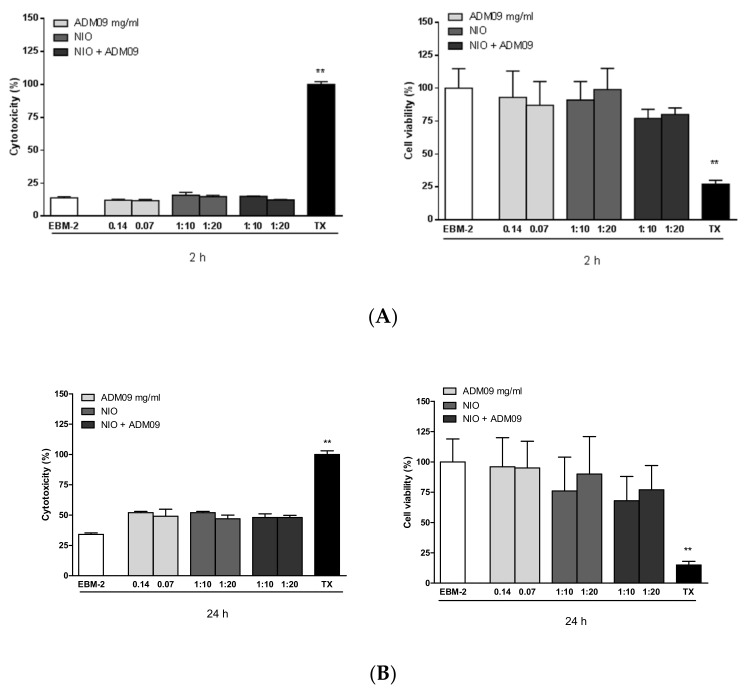
Citotoxicity studies at different concentrations of ADM_09 (0.14 and 0.07 mg/mL), empty noisomes (NIO), and loaded niosomes (NIO ADM_09) for 2h (**A**) and 24 h (**B**). ** *p* < 0.05.

**Figure 7 pharmaceutics-11-00669-f007:**
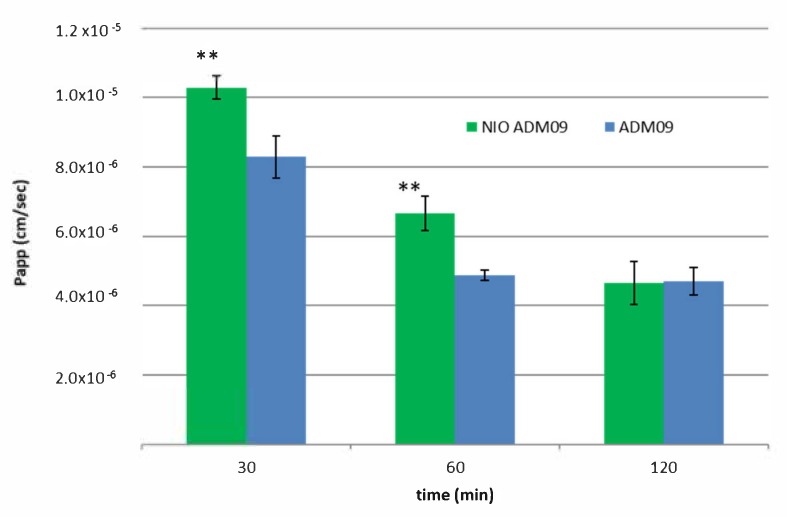
hCMEC/D3 apparent permeability coefficient (Papp) of free drug (ADM_09 blue) and loaded in niosomes (NIO ADM_09 green) after incubation of 30, 60, and 120 min. ** *p* < 0.05.

**Table 1 pharmaceutics-11-00669-t001:** Effect of different preparation methods: thin layer evaporation–paddle (TLE-P), thin layer evaporation–vortex (TLE-V) and thin layer evaporation–frozen and thawed (TLE-F) on niosomes (NIO) particle size (p.s.), ζ-potential (ζ.pot.), and polydispersion index (PDI).

Preparation Method	p.s. (nm ± s.d.)	PDI ± s.d.	ζ.pot. (mV ± s.d.)
TLE-P	135.37 ± 2.22	0.21 ± 0.01	−15.57 ± 0.35
TLE-V	235.97 ± 5.33	0.35 ± 0.07	−22.57 ± 0.81
TLE-F	131.17 ± 4.22	0.20 ± 0.02	−16.40 ± 0.60

**Table 2 pharmaceutics-11-00669-t002:** Effect of different component composition in mg/mL on NIO particle size (p.s.), ζ-potential (ζ.pot.), and polydispersion index (PDI).

Batch	Span 60	Span 40	CHL	CHE	SOL	HPβCD	p.s. (nm ± s.d.)	PDI ± s.d.	ζ.pot. (mV ± s.d.)
0	8.00		5.73		5.33		135.37 ± 2.22	0.21 ± 0.01	−15.57 ± 0.35
1	5.73		9.53		3.81		418.47 ± 6.29	0.31 ± 0.04	−22.40 ± 1.32
2	11.43		4.75		2.86		362.97 ± 2.22	0.32 ± 0.04	−27.30 ± 2.38
3	13.34		2.86		2.86		301.70 ± 4.11	0.25 ± 0.02	−21.30 ± 1.78
4		8.00					227.07 ± 3.63	0.29 ± 0.01	−18.77 ± 0.78
5				5.73			214.37 ± 6.28	0.32 ± 0.04	−21.10 ± 1.15
6		8.00		5.73			189.93 ± 3.76	0.26 ± 0.01	−20.43 ± 0.64
7	8.00		5.73		5.33	4.88	159.17 ± 0.55	0.19 ± 0.01	−19.40 ± 2.31

**Table 3 pharmaceutics-11-00669-t003:** Effect of different composition on NIO ADM09 drug loading (DL%), entrapment efficiency (EE%), particle size (p.s.), ζ-potential (ζ.pot.), and polydispersion index (PDI).

Batch	DL%	EE%	p.s. (nm ± s.d.)	PDI ± s.d.	ζ.pot. (mV ± s.d.)
0.1	0.17	27.07	192.33 ± 8.69	0.27 ± 0.02	−16.60 ± 0.60
6.1	0.06	3.64	227.70 ± 2.21	0.26 ± 0.01	−19.90 ± 0.26
7.1	0.02	2.50	224.67 ± 1.81	0.26 ± 0.03	−20.70 ± 0.95

**Table 4 pharmaceutics-11-00669-t004:** Stability studies in fetal bovine serum: effect on empty and loaded NIO particle size (p.s.) and polydispersion index (PDI).

Time (min)	NIO		NIO ADM_09	
	p.s. (nm ± s.d.)	PDI ± s.d.	p.s. (nm ± s.d.)	PDI ± s.d.
0	145.37 ± 2.22	0.31 ± 0.01	202.12 ± 4.23	0.37 ± 0.02
30	146.41 ± 0.42	0.32 ± 0.01	212.31 ± 5.29	0.41 ± 0.12
60	145.35 ± 1.73	0.32 ± 0.01	215.58 ± 3.15	0.37 ± 0.18
90	144.24 ± 3.61	0.34 ± 0.05	216.23 ± 6.19	0.36 ± 0.21
120	141.68 ± 2.71	0.39 ± 0.07	208.41 ± 4.12	0.32 ± 0.09
150	143.67 ± 3.72	0.37 ± 0.07	225.33 ± 2.89	0.35 ± 0.19
180	148.65 ± 3.09	0.33 ± 0.04	219.19 ± 5.74	0.38 ± 0.15

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
