# Peer review of "Niosomal Formulation of a Lipoyl-Carnosine Derivative Targeting TRPA1 Channels in Brain"

_pharmaceutics, 2019, doi:10.3390/pharmaceutics11120669_

Round 1
Reviewer 1 Report
Reviewer comment (Pharmaceutics-656725).
In this study, the authors investigated the activity of ADM_09, a recently developed lipoic acid-based TRPA1 antagonist with anti-oxidative effect, on neocortex cultures was studied and an efficient niosomal formulation to cross the blood-brain barrier in vitro was tried to be developed with the aim of increasing the concentration of ADM_09 into the brain and selectively deliver it to the CNS rapidly after parenteral administration. The ADM_09 showed rapid direct effect on neuronal network excitability. Also, the NPG-functionalized niosomal formulation incorporating ADM_09 could efficiently pass through the in vitro BBB model, the layer of hCMEC/D3 cells cultured onto transwell plate. Based on these results, the niosomal formulation of ADM_09 may be useful formulation for the treatment of oxaliplatin-induced neuropathic pain and inflammatory trigeminal allodynia. However, several concerns should be addressed before publication in Pharmaceutics, as follows.
1) Regarding the niosomal formulation, the amount of NPG modification was fixed. How did the authors decide the amounts? Please discuss about that point.
2) In figure 7, the authors showed the results of BBB permeability studies, in which NPG-functionalized niosomes had been employed. How about the effect of GLUT-1 inhibition or glucose competition on the formulation permeability?
3) The authors prepared niosomes having 200 nm, with less than PDI<0.2, presumably for in vivo application. Moreover, NPG modification was performed to increase BBB permeation. How about the concentration of ADM_09 in the brain of animals after parenteral administration? At least, the authors should indicate blood concentration of ADM_09, and stability of the niosomes in the presence of serum.
4) The authors should investigate the effect of ADM_09 incorporated in niosomes. The authors examined its effect by using only free form of ADM_09 onto neurons. Also, if possible, it’s better to examine the effect in transwell system.
Author Response
The Authors thanks the Reviewer for the constructive and stimulant comments. Please find below the point-by-point responses to the comments.
Q.1) Regarding the niosomal formulation, the amount of NPG modification was fixed. How did the authors decide the amounts? Please discuss about that point
A.1) Thank you for the question: NPG content was fixed since an increase of concentration resulted in a NPG crystallization (Bragagni et al., 2012). The comment was added in the text.
Q.2) In figure 7, the authors showed the results of BBB permeability studies, in which NPG-functionalized niosomes had been employed. How about the effect of GLUT-1 inhibition or glucose competition on the formulation permeability?
A.2) The Authors thank for the question. Further studies are needed to study the mechanism of NPG interaction and NPG-niosomes effect. Unfortunately, such kind of study requires more than 10 days to be performed and could not describe the mechanism of action of such formulation. In fact, the NPG- niosomes represent a vehicle to let the drug overcome the BBB that could simply increase the drug concentration near to the BBB and promote the drug permeation. What is certain is that NPG was successfully employed for brain targeting of the vasoactive intestinal peptide (Dufes et al., 2004) doxorubicin (Bragagni et al. 2012) and dynorphin-B (Bragagni et al., 2014). Although the exact mechanism of action needs further investigation, the recognition of glucosamine exposed on NPG-bearing niosomes by the glucose transported GLUT-1, highly expressed on the BBB cell, has been supposed (Dufes et al., 2004). The comment and the reference were added in the text.
Q.3) The authors prepared niosomes having 200 nm, with less than PDI<0.2, presumably for in vivo application. Moreover, NPG modification was performed to increase BBB permeation. How about the concentration of ADM_09 in the brain of animals after parenteral administration? At least, the authors should indicate blood concentration of ADM_09, and stability of the niosomes in the presence of serum.
A.3) The review comment is correct: such kind of formulation is aimed for a parenteral administration. At the moment the final formulation wasn’t tested on animals so we haven’t any data about blood or brain concentration. In-vivo studies will be certain performed but, in these pre-formulation studies, the use of in-vitro can be useful to reduce time, cost and avoid animal suffering. On the other hand, thanks to the reviewer suggestion, stability studies of niosomes in fetal bovine serum were performed, in order to better simulate the in-vivo behaviour. Even if the PDI of niosomes lightly increase, probably due to the protein presence, particle size was maintained during the first 3 hours thus confirming niosomes stability in the medium. The comments were added in the text.
Q.4) The authors should investigate the effect of ADM_09 incorporated in niosomes. The authors examined its effect by using only free form of ADM_09 onto neurons. Also, if possible, it’s better to examine the effect in transwell system.
A.4) We tested on neurons only the effect of free ADM_09 because our niosomes are merely a vehicle to facilitate the drug’s crossing the BBB. Further studies are needed to determine the mechanism of NPG interaction and the NPG-niosomes effect. Nonetheless, at the moment we have no evidence that this kind of niosomes can cross intact the BBB. Transwell systems are not applicable to the currently available MEA systems.
Reviewer 2 Report
The manuscript describes the preparation and use of a niosomal formulation to deliver lipoyl-carnosine that can act as antioxidant and TRP1 antagonist to reduce pain and inflammation. The main objective of the research is to develop a formulation with the possibility of crossing the Blood Brain Barrier (BBB) to deliver the drug into the brain. The synthesized niosomes were homogenous, stable up to 8 months at 5ºC and they were able to release the cargo without apparent toxicity. In addition niosomes have an increased permeability in a cell model of human BBB. On the negative site, the drug loading and entrapment efficiency are relatively low and the drug release is relatively too fast. Results are interesting. There are a few minor suggestions to
Table 2, there are a few abbreviations that are not defined such as SOL ?, HP C D?
Figure 6. Abbreviations EMB-2 and TX are not defined
Reference 38, please add volume and page number. Reference 27 has no page number.
Some references lacks separation between year and volume (see for example ref 32).
Check format of references. Some have “:” as separation between volume and page (ref. 32, 25), and some others have “,”. Some have a point at the abbreviation, others have no point.
Minor grammatical errors:
Page 7, line 298: R-a-lipoic acid, “a” should be alpha
Page 12, line 383, Figure 6 legend: Citotoxicity should be Cytotoxicity
Author Response
The Authors thanks the Reviewer for the constructive and stimulant comments. Please find below the point-by-point responses to the comments.
Reviewer comments highlights some defects in our formulation that should be absolutely improved. Some studies, aimed to improve drug entrapment and control drug release are now in progress and will be discussed in a future paper.
Q. 1) Table 2, there are a few abbreviations that are not defined such as SOL ?, HP C D?
Figure 6. Abbreviations EMB-2 and TX are not defined
A. 1) The missing abbreviations were added in the text
Q.2) Reference 38, please add volume and page number. Reference 27 has no page number.
Some references lacks separation between year and volume (see for example ref 32).
Check format of references. Some have “:” as separation between volume and page (ref. 32, 25), and some others have “,”. Some have a point at the abbreviation, others have no point.
A.2) References missing data and format have been checked and revised
Q. 3) Minor grammatical errors:
Page 7, line 298: R-a-lipoic acid, “a” should be alpha
Page 12, line 383, Figure 6 legend: Citotoxicity should be Cytotoxicity
A.3) The error were corrected
All the revisions are highlighted using the "Track Changes" function in Microsoft Word.
Round 2
Reviewer 1 Report
The revised manuscript was improved before first submission.However, the reviewer ask the authors to address the following point.
1) The authors examined the stability of the prepared niosomes in the presence of serum. However, the results were not shown in the revised manuscript, only sentenses. The data of particle size and PDI should be shown with statistical analysis.
Author Response
Q.1) The authors examined the stability of the prepared niosomes in the presence of serum. However, the results were not shown in the revised manuscript, only sentenses. The data of particle size and PDI should be shown with statistical analysis.
A.1) The Reviewer is right: the data are reported in the text summarized as table 4. The statistical analysis conducted ad usual evidenced a significative difference just respect to the niosomes in PBS at the same temperature, as reported in the text.